# *auts2* Features and Expression Are Highly Conserved during Evolution Despite Different Evolutionary Fates Following Whole Genome Duplication

**DOI:** 10.3390/cells11172694

**Published:** 2022-08-30

**Authors:** Constance Merdrignac, Antoine Emile Clément, Jérôme Montfort, Florent Murat, Julien Bobe

**Affiliations:** INRAE, LPGP UR1037, Fish Physiology and Genomics, Campus de Beaulieu, F-35000 Rennes, France

**Keywords:** autism spectrum disorders (ASDs), neurodevelopmental disorders (NDDs), brain, evolution, teleost, medaka, zebrafish

## Abstract

The *AUTS2* gene plays major roles during brain development and is associated with various neuropathologies including autism. Data in non-mammalian species are scarce, and the aim of our study was to provide a comprehensive analysis of *auts2* evolution in teleost fish, which are widely used for in vivo functional analysis and biomedical purposes. Comparative genomics in 78 species showed that *auts2a* and *auts2b* originate from the teleost-specific whole genome duplication (TGD). *auts2a,* which is highly similar to human AUTS2, was almost systematically retained following TGD. In contrast, *auts2b,* which encodes for a shorter protein similar to a short human AUTS2 isoform, was lost more frequently and independently during evolution. RNA-seq analysis in 10 species revealed a highly conserved profile with predominant expression of both genes in the embryo, brain, and gonads. Based on protein length, conserved domains, and expression profiles, we speculate that the long human isoform functions were retained by *auts2a*, while the short isoform functions were retained by *auts2a* and/or *auts2b*, depending on the lineage/species. *auts2a* showed a burst in expression during medaka brain formation, where it was expressed in areas of the brain associated with neurodevelopmental disorders. Together, our data suggest a strong conservation of *auts2* functions in vertebrates despite different evolutionary scenarios in teleosts.

## 1. Introduction

The *AUTS2* gene (activator of transcription and developmental regulator, previously known as autism susceptibility candidate 2) is associated with multiple neurodevelopmental disorders (NDDs) and neurological disorders (see [1,2,3,4] for a recent review). *AUTS2* was originally linked to autism spectrum disorders (ASDs) [5] and subsequently associated with other intellectual disabilities, short stature, microcephaly, cerebral palsy, facial dimorphism [6], developmental delays [7], mental retardation [8], speech and language disorders [9], addictive behavior [10], and schizophrenia [11]. The human *AUTS2* gene encodes a full-length protein of 1259 amino acids [6]. While numerous splice variants have been described, the best documented variants encode for a 786-amino-acid (S-AUTS2-Var1) protein and a 711-amino-acid (S-AUTS2-Var2) protein, both corresponding to the C-terminal portion of the full-length protein [2,4,6]. In humans, *AUTS2* is expressed in various tissues including the fetal brain [5]. In mice, *Auts2* is expressed in the developing brain, where it localizes to the developing cortex, cerebellum, and hippocampus [12]. In the nucleus, AUTS2 is believed to activate the transcription of genes involved in brain development and NDDs, many of which being expressed in the developing mouse forebrain [13]. It was also shown that AUTS2 interacts with an epigenetic regulator type I Polycomb repressive complex (PRC1) [14]. It was however hypothesized that AUTS2 could function as a transcriptional activator or repressor depending on the cellular context [2,14,15]. In the nucleus, AUTS2 was also hypothesized to interact with RNA, either directly or through RNA binding proteins to ultimately regulate neuronal differentiation [4,15,16]. Finally, *Auts2* is also able to regulate cytoskeletal dynamics in the cytoplasm to ultimately regulate neurite outgrowth and branch formation, as well as neuronal formation in mice [17]. In mice, *Auts2* invalidation resulted in neonatal death, as shown by several independent studies specifically targeting different exons [16,17]. When heterozygous mutants were used or conditional knock-out was carried out, abnormal behavioral phenotypes were observed in independent studies targeting different exons [16,17,18,19,20]. While *Auts2* has been studied in humans and mice, existing data in other mammalian species are scarce, with a single study conducted in pig [21]. To the best of our knowledge, *Auts2* orthologs were never studied in any non-mammalian tetrapod species. In teleost fish, *auts2* was studied in zebrafish (*Danio rerio*), in which two distinct genes, *auts2a* and *auts2b*, were reported. Consistent with existing data in mice, *auts2a* and *auts2b* are expressed in the developing zebrafish brain [22]. Knock-down of *auts2a* in zebrafish resulted in phenotypes that were globally consistent with murine data, even though behavioral phenotypes were not assessed [6,23]. In rainbow trout (*Oncorhynchus mykiss*), a maternal temperature stress resulted in impaired behavior in the progeny and dysregulation of several neurodevelopmental genes in the eggs, including a reduced expression of *auts2* [24]. Despite the existence of two *auts2* genes in zebrafish, it remains unclear if these genes originate from the whole genome duplication (WGD) event that occurred at the root of teleosts (i.e., the teleost-specific genome duplication (TGD). It is also unclear if the presence of two *auts2* genes is common to all teleosts or limited to several species such as zebrafish. The existence of two *auts2* gene copies in zebrafish, and possibly other teleost species, raises the question of the evolution of different functions for *auts2a* and *auts2b* after WGD. In this context, the aim of the present study was to comprehensively characterize the evolutionary history of *auts2* genes in teleosts using comparative genomics and shed new light on the evolution and/or conservation of the expression of *auts2* genes in teleost fish. This information is currently missing and required to further investigate the functions of *auts2* using popular teleost fish models to better understand the role(s) and function(s) of *auts2* during neurodevelopment in a biomedical context. Here, we show that most teleost species retained only the *auts2a* copy, which exhibits highly conserved protein size and domain features with humans and mice. Using Japanese medaka (*Oryzias latipes*), a model species for neurodevelopment in which only *auts2a* was retained, we show that the spatio-temporal gene expression is also highly conserved with expression in areas of the brain that are associated with neurodevelopmental disorders.

## 2. Materials and Methods

### 2.1. Sequence Availability Synteny Analysis

The synteny analysis was performed using gene orthology information available in genomicus version 03.01 (https://www.genomicus.bio.ens.psl.eu/genomicus-fish-03.01/cgi-bin/search.pl, accessed on 28 March 2022). For all species, *auts2* accession numbers are available in Appendix A. For Atlantic salmon (*Salmo salar*), rainbow trout (*Oncorhynchus mykiss*), and coho salmon (*Oncorhynchus kisutch*), gene synteny was analyzed using Ensembl v106 (https://uswest.ensembl.org/index.html?redirect=no, accessed on 20 April 2022). For allis shad (*Alosa alosa*), the synteny analysis was performed using a BLAST search to identify genes present in conserved synteny blocks in the vicinity of *auts2a* and *auts2b* on the publicly available genome assembly (Accession #: GCA_017591415.1). For European conger (*Conger conger*), we used the genome assembly available in the omics Dataverse303 (open-source research data repository) server (https://doi.org/10.15454/GWL0GP, accessed on 5 April 2022) [25] to perform a similar analysis. Genome assembly versions, gene positions on the genome, and corresponding accession numbers are available in Appendix A.

### 2.2. Gene Remnants’ Identification

To identify potential gene remnants, we performed TBLASTN analysis. The AUTS2 family conserved protein domain was blasted against genomic sequences located in relevant syntenic block possibly bearing *auts2a* or *auts2b* gene remnants. To identify *auts2b* gene remnants, we used zebrafish *auts2b* exon 9 as the bait. To identify *auts2a* gene remnants, we used red-bellied piranha (*Pygocentrus nattereri*) *auts2a* exon 7 as the bait for Mexican tetra (*Astyanax mexicanus*) and electric eel (*Electrophorus electricus*) and *auts2a* exon 10 as the bait for channel catfish (*Ictalurus punctatus*) and iridescent shark catfish (*Pangasianodon hypophthalmus*).

### 2.3. Phylogenetic Analysis

Phylogenetic analysis was performed using Auts2 full-length protein sequences of 33 species with Molecular Evolutionary Genetics Analysis (MEGA) version 11.0.11 software (State College, PA, USA). Sequences were first aligned using CLUSTALW, and phylogenetic trees were generated using the maximum likelihood method (based on the Whelan and Goldman + Freq. model) with 500 bootstrap replicates and subsequently confirmed using the neighbor-joining and minimum likelihood methods (500 bootstrap replicates).

### 2.4. Read Processing and Estimation of Expression Levels

RNA-seq datasets across 11 tissues from 10 species from a previous study [26] were downloaded from SRA (Accession #: SRR1524271-81 (Japanese medaka), SRR2045415-25 (Atlantic cod), SRR1533651-61 (Northern pike), SRR1524238-49 (zebrafish), SRR2045404-14 (Mexican tetra Pachon cave morph), SRR2045426-36 (Mexican tetra surface morph), SRR1533640-50 (iridescent shark catfish), SRR1532799-809 (allis shad), SRR1532756-65 (European eel), SRR1524250-60 (spotted gar), SRR1524261-70 (bowfin); see Appendix A). Raw reads with known 3′ adaptor and low-quality bases (Phred score < 20) were trimmed with TrimGalore (version 0.6.6, Cambridge, UK) (https://github.com/FelixKrueger/TrimGalore, accessed on 28 April 2022) (parameters: -r_clip 13 -three_prime_clip 2). Next, we mapped the trimmed reads from each library against the reference genome and annotated transcripts of the species (Appendix A) using Salmon (version 1.8) [27] (parameters: defaults). Gene-expression levels were measured in transcripts per kilobase million (TPM), a unit which corrects for both feature length and sequencing depth.

### 2.5. Sequence Features

To investigate conserved sequence features between species, amino acid sequences (accession numbers available in Appendix A) were aligned using the BioEdit software (version 7.0.5.3, Raleigh, NC, USA) and conserved domains were visually identified based on existing data [4].

### 2.6. Embryo Sampling

Fertilized medaka eggs were collected, and developing embryos were subsequently incubated at 27 °C in mineral medium (water, 0.1% NaCl, 0.003% KCl, 0.004% CaCl_2_, 0.016% MgSO_4_) until they reached the targeted developmental stage. Embryos were then snap frozen in liquid nitrogen and stored at −80 °C until RNA extraction. The following eleven development stages were used to characterize the *auts2a* expression profile throughout medaka embryonic development: 1, 4, 8–9, 11, 17–18, 20, 25, 29, 30, 36, 38, according to Iwamatsu [28]. For each developmental stage, 4 biological replicates, corresponding to pools of embryos, were used. At early stages (stages 1 to 11), pools of 60–95 embryos were used. For later stages (stage 17–18 to 38), pools of 28–44 embryos were used.

### 2.7. RNA Extraction

Frozen medaka embryos were lysed with a Precellys Evolution Homogenizer (Ozyme, Bertin Technologies, Montigny-le-Bretonneux, France) in TRI Reagent (TR118; Euromedex, Cincinnati, OH, USA), and total RNA was extracted according to the manufacturer’s instructions. After extraction, RNA concentration was measured using a nanodrop, and luciferase exogenous RNA was added in each sample at a ratio of 9 pg per embryo for subsequent qPCR data normalization. RNA was stored at −80 °C until reverse transcription.

### 2.8. Reverse Transcription and Quantitative PCR

Reverse transcription (RT) was performed from 1μg of total RNA using the Maxima First Strand cDNA Synthesis Kit for RT-qPCR (Thermo Scientific, Waltham, MA, USA) following the manufacturer’s instructions. Real-time PCR reactions were performed using the PowerUp SYBR Green Master Mix (Applied Biosystems, Waltham, MA, USA) according to the manufacturer’s instructions in a total volume of 10 μL, containing RT products diluted at 1:62.5 and 400 nM of each primer. Primer sequences are available in Appendix A. As *auts2a* is believed to have multiple transcription start sites and transcripts [22], we used primers surrounding the exons 2–3 junction to target the full-length isoform and the exons 18–19 junction to target both full-length and short C-terminal isoforms. Each biological replicate was analyzed in quadruplicate. The qPCR cycle was run on the Light Cycler 480 (Roche, Bâle, Switzerland) and included a pre-amplification step of 2 min at 50 °C and 2 min of initial denaturation at 95 °C, followed by 40 cycles at 95 °C for 3 sec and 60 °C for 30 sec. To visualize the melting curve, a third step included 15 sec of denaturation at 95 °C followed by 1 sec at 60 °C and a final denaturation at 95 °C. Finally, a cooling step was performed for 30 sec at 40 °C. The relative abundance (absorbance quantification/fit points) of target cDNA was evaluated from a serially diluted cDNA pool (standard curve) using the Light Cycler 480 software (version 1.5.1.62, Roche, Bâle, Switzerland). Before analysis, gene expression data were normalized using luciferase signal.

### 2.9. Statistical Analysis

To compare *auts2a* expression levels between stages for each set of primers independently, a Kruskal test was performed (*p* < 0.05), and a pairwise Wilcoxon signed rank test was subsequently used (*p* < 0.05) to make two by two comparisons between stages.

### 2.10. RNAscope

RNAscope is a state-of-the-art in situ hybridization method [29,30,31]. It was carried out following the manufacturer’s instructions (ACDBio, bio-techne, Newark, CA, USA) with minor changes inspired by previous studies conducted on whole-mount tissues [30,31]. 

#### 2.10.1. Probe Design

The RNAscope *auts2a* probe was designed by ACDBio company. It consisted of a 20 ZZ probes targeting nucleotides 3356–4345 of the Japanese medaka *auts2a* full-length transcript (XM_011483791.3).

#### 2.10.2. Embryo Sampling and Preprocessing

Stage 29 embryos were fixed in 4% paraformaldehyde (PFA)/phosphate-buffered saline (PBS: 137 mM NaCl, 2.7 mM KCl, 10 mM Na2HPO4, 1.8 mM KH2PO4, pH 7) one day at room temperature. The PFA fixation was stopped by a 10 min PBS + 0.1% tween (PBST) wash. The chorion, a thick acellular envelope surrounding the egg, was removed manually using thin forceps. Embryos were then gradually dehydrated in methanol/PBST (increasing methanol concentration: 10 min in 25% methanol, 10 min in 50% methanol, 10 min in 75% methanol, 10 min in 100% methanol) and were subsequently stored in 100% methanol at −20 °C until the RNAscope assay was performed. Assays were run using five embryos for each condition. Embryos were put in mesh inserts within cryotubes and culture plate wells. RNAscope reagent steps (Protease Plus, AMP, HRP-C1/HRP-C3, Opal520 and HRP blocker steps) were carried out in 2 mL round-bottom plastic cryotubes and other steps in a 24-well cell culture plate (working volume: 750 μL).

#### 2.10.3. Sample Pre-Treatment 

First, samples were gradually rehydrated in methanol/PBST (decreasing methanol concentration: 10 min in 75% methanol, 10 min in 50% methanol, 10 min in 25% methanol, 10 min in 100% PBST). Unmasking was then performed using the RNAscope Target Retrieval reagent (ACDBio, ref: 322000). The aim of this step was to improve the signal quality. Samples were incubated (15 min at 100 °C) in Target Retrieval reagent 1X previously briefly boiled in a microwave oven. Two 10 min PBST washes were then performed. Embryos were then permeabilized using a few drops of RNAscope Protease Plus Reagent (ACDBio, ref: 322331) for 15 min at 40 °C. Protease Plus action was stopped by three 10 min PBST washes.

#### 2.10.4. RNAscope Multiplex Fluorescent V2 Assay 

The assay was carried out using the RNAscope Multiplex Fluorescent V2 kit (ACDBio, ref: 323100). The RNAscope 3-plex negative control channel 3 probe (ACDBio, ref: 320871), targeting the *Bacillus subtilis dapB* gene (accession # EF191515), was used as a negative control. Hybridization was carried out using a few drops of pre-warmed RNAscope probe, overnight at 50 °C. The following day, three 10 min 0.2X saline-sodium citrate + 0.01% tween (0.2X SSCT) washes were performed. Embryos were fixed with 4% PFA in PBST for 6 min. The fixation was followed by three 10 min 0.2X SSCT washes. Then, six steps were carried out to hybridize preamplifiers and amplifiers, each one with a different RNAscope reagent. Each step started with adding a few drops of RNAscope reagent on the sample at 40 °C and lasted as follows: AMP1: 45 min, AMP2: 45 min, AMP3: 30 min, HRP-C1/HRP-C3: 45 min. Between each RNAscope reagent incubation, three 10 min 0.2X SSCT washes were performed at room temperature. RNAscope HRP reagent was used according to the probe channel (HRP-C1 for RNAscope channel 1 probes, HRP-C3 for RNAscope 3-plex negative control channel 3 probe). Finally, the signal was revealed using Fluorescein Opal520 reagent (1/1500 dilution of Fluorescein Opal520 reagent (Akoya Biosciences, Marlborough, MA, USA, ref: FP1487001KT) in RNAscope TSA buffer). The revelation was conducted in the dark for 30 min. Then, samples were washed three times 10 min in 0.2X SSCT in the dark. Subsequently, the HRP blocker step was carried out in the dark, and samples were washed three times for 10 min in 0.2X SSCT in the dark. Finally, embryos were incubated with methyl green overnight at 4 °C in the dark.

#### 2.10.5. Microscopy

Images of whole embryos were acquired on a TCS SP8 laser scanning confocal microscope with a 25X objective (0.95 NA) using the LASX software (Leica Microsystems Wetzlar, Germany, version 3.5.7). Embryos were orientated and immobilized in 2% agarose grooves made using Stampwell molds (Idylle, Cape Town, South Africa) prior to observation. Maximum Z projection images of RNAscope staining and standard deviation Z projection images of methyl green staining were obtained using the ImageJ (FIJI) software (version 1.53f51, Bethesda, MD, USA).

## 3. Results

### 3.1. Evolution of the auts2 Gene Repertoire

The analysis of the genomic region in the vicinity of the *Auts2* gene did not show any long-range conservation between tetrapods and teleosts (Figure 1A and Appendix A). We used the spotted gar (*Lepisosteus oculatus*), a species belonging to the Holostei group, which did not experience the teleost-specific whole genome duplication (TGD), to bridge the gap between tetrapods and teleosts [32]. We observed a conserved gene synteny (i.e., conservation of gene blocks on chromosome regions) between tetrapods and spotted gar and between spotted gar and teleosts (Figure 1A and Appendix A). The synteny was especially well conserved among teleosts, with several orthologous genes such as *med13*, *vmp1*, *ctlc*, and *map6* present in the vicinity of *auts2* genes (Figure 1A and Appendix A). The conserved synteny between spotted gar and teleosts, in which two distinct genomic regions could systematically be identified, indicates that teleost *auts2a* and *auts2b* originate from the TGD. This observation was further supported using bowfin (*Amia calva*), another Holostei species (Appendix A). When performing phylogenetic analysis, we clearly observed a separation between *auts2a* genes and *auts2b* genes (Figure 1B). A similar tree topology was observed using three different tree-building methods (Appendix A).

The analysis of the *auts2* gene repertoire in teleost species revealed a striking diversity in gene retention patterns among species. Two copies of the gene, *auts2a* and *auts2b*, were found in European conger (*Conger conger*), zebrafish, and Northern pike (*Esox lucius*) (Figure 1A and Appendix A). In contrast, only *auts2a* was found in European eel (*Anguilla anguilla*), Asian arowana (*Scleropages formosus*), Atlantic cod (*Gadus morhua*), and Japanese medaka, while only *auts2b* could be identified in the two morphs of Mexican tetra (*Astyanax mexicanus*) (Figure 1A and Appendix A). We then performed a more comprehensive analysis using 76 teleost species. We observed that only *auts2a* could be identified in 52 species belonging to Acanthomorphata, a large group of fish species that encompasses medakas, guppies, mollies, stickleback, perch, flatfishes, tetraodons, and cods (Figure 2A and Figure 3). Non-functional remnants of *auts2b* genes could systematically be found in ohnologous regions when the *auts2b* exon 9 amino acid sequence was blasted onto the genome (Figure 1A and Appendix A) in all species for which the analysis was performed (*n* = 8). This demonstrates that *auts2b* was lost in this lineage, which diverged approximately 150 Mya [33]. Similarly, *auts2b* was lost in the three Osteoglossomorpha species (*Scleropages formosus*, *Paramormyrops kingsleyae*, and *Arapaima gigas*) that we investigated (Figure 2A and Figure 3). In these species, we did not succeed in identifying non-functional *aust2b* gene remnants in available genome assemblies. This absence of *auts2b* gene remnants would be consistent with a rapid loss of the gene after the Osteoglossomorpha radiation, which occurred approximately 200–250 Mya [33]. In contrast, a great diversity of evolutionary fates was observed in the Otomorpha clade (Figure 2A and Figure 3). While only *auts2a* was retained in Atlantic herring (*Clupea harengus*), other species retained either both genes, in the case of electric eel (*Electrophorus electricus*), red-bellied piranha (*Pygocentrus nattereri*), and zebrafish, or only *auts2b*, in the case of Mexican tetra (both surface and Pachon cave morphs), channel catfish (*Ictalurus punctatus*), and iridescent shark catfish (*Pangasianodon hypophtalmus*). These observations were supported by the identification of possible gene remnants spanning over 6–8 amino acids, depending on the species (Appendix A), corresponding to a highly conserved sequence of the Auts2 family [22] in the *auts2a* syntenic block.

Together, these observations suggest relatively recent species-specific gene losses in Otomorpha resulting in highly variable *auts2* gene repertoires in this group. This recent species-specific loss was also observed in Elopomorpha in which *auts2b* was found in European conger, but not in European eel (Figure 2A, Figure 3 and Appendix A), despite the short divergence time (i.e., approximately 90 Mya) between the two species [33]. The loss of *auts2b* in European eel was further supported by the presence of a possible gene remnant spanning over six amino acids (Appendix A), corresponding to a highly conserved sequence of the Auts2 family [22] in the *auts2b* syntenic block.

In summary (Figure 3), the comprehensive gene analysis shows that, in contrast to many other genes [34,35] including *foxl3* [36], *nmp2a*, *npm2b* [37], *shbga*, *shbgb*, *pros1*, and *gas6* [38], to name just a few, a rapid loss of one of the two *auts2* gene copies did not occur after TGD. We however observed the loss of *auts2b* in the Acanthomorphata ancestor, approximately 150 Mya. Similarly, *auts2b* was not present in Osteoglossomorpha, suggesting that the loss of the gene could have occurred between 200 and 250 Mya [33]. In contrast, we observed much more recent losses of either *auts2a* or *auts2b*, which occurred in a species-specific manner in different clades. We observed that additional WGD events that occurred in Salmonidae [39] and Cyprininae [40] also impacted the *auts2* gene repertoires. In Cyprininae, both genes were retained in two copies (*auts2a1*, *auts2a2*, *auts2b1* and *auts2b2*) in most species (Figure 2 and Appendix A). We also observed additional copies of *auts2a* and *auts2b* in goldfish and carp, respectively. It remains unclear if these additional copies are due to local duplications or to genome assembly artefacts (Figure 2 and Appendix A). Similar observations were made in Salmonidae, in which two copies of *auts2a*, *auts2a1*, and *auts2a2*, could be identified in rainbow trout (*Oncorhynchus mykiss*), coho salmon (*Oncorhynchus kisutch*), Atlantic salmon (*Salmo salar*), and Arctic char (*Salvelinus alpinus*). In contrast, a single copy of *auts2b* was retained in these three species (Figure 2 and Appendix A). 

### 3.2. Auts2 Protein and Gene Sequence Features

The human *AUTS2* gene is very large and consists of 19 exons [5]. This exon structure is conserved in mice [12], zebrafish [6,22], and medaka (Figure 4), suggesting a highly conserved genomic structure among vertebrates. The coding sequence, however, starts on exon 2 in zebrafish [22] and medaka (Figure 4), while it starts on exon 1 in humans [5] and mice [12]. The full-length AUTS2 protein was extensively described in humans [5] and mice [12]. The human AUTS2 includes three nuclear localization sequence (NLS) motifs located in the N-terminal region of the protein (Figure 4). Two proline-rich repeat domains, PR1 and PR2, are encoded by exons 7–8 and 9–13, respectively (Figure 4). The PPPY and HXR domains are located between the two PR domains and are encoded by exon 9. A serine-rich repeat domain (SRR) is encoded by exon 7. Finally, an AUTS2 domain, that is also found in all members of the Auts2 family (i.e., *AUTS2*, *FBRS*, and *FBRSL1*) is encoded by exons 13–19. All these domains are also present in zebrafish [6] and medaka (Figure 4) Auts2a, indicating a highly conserved protein structure in vertebrates. We however observed differences in the size of the HXR domain with 11 HX repeats in zebrafish and medaka Auts2a and 9 HX repeats in human AUTS2. When comparing Auts2a and Auts2b, we observed that the HXR domain was shorter in Auts2b in comparison to Auts2a (Figure 4 and Appendix A).

### 3.3. Evolution of auts2 Gene Expression

RNA-seq data available from the PhyloFish database [26] were remapped on available genomes to characterize *auts2a* and *auts2b* expression in different tissues and organs, as well as embryos. We observed highly conserved expression patterns with a predominant expression in the embryo, adult brain, and gonads. This pattern was strikingly similar in species in which either *auts2a* or *auts2b* was retained. In zebrafish, in which both *auts2a* and *auts2b* are present, both genes exhibited a similar expression pattern both in terms of tissue distribution and expression levels with a predominant expression in the embryo, adult brain, and testis. In contrast, *auts2b* expression in Northern pike appeared to be extremely reduced in comparison to *auts2a*, suggesting a possible pseudogenization of this copy (Figure 2B).

### 3.4. Expression of auts2a during Japanese Medaka Neurodevelopment

Using medaka as a model, we monitored *auts2a* expression throughout embryonic development with special interest in embryogenic brain formation. Given the existence of different transcripts, we used quantitative RT-PCR to target the N-terminal region (exons 2–3) or the C-terminal region (exons 18–19) of the protein. We were able to detect *auts2a* above background levels in fertilized eggs and at various developmental stages prior to zygotic genome activation (Figure 5B), indicating that *auts2a* mRNA is maternally inherited. Similar profiles were observed when exons 2–3 and 18–19 were targeted. A burst in *auts2a* expression was subsequently observed with a 7-fold increase at the early neurula stage (stage 17 according to Iwamatsu [28]) for both targeted regions. Throughout neurulation (stages 17–24 according to Iwamatsu [28]), *auts2a* expression exhibited a stepwise increase at stages 20 and 25 (i.e., neural tube development) when targeting exons 2–3. This increase was however not observed until stage 25 when targeting exons 18–19. We subsequently observed, for both targeted regions, a dramatic increase in *auts2a* expression at stage 29, which corresponds to the beginning of late embryonic brain establishment. Expression further increased at stage 36 and remained elevated during subsequent embryo development (i.e., stage 38) (Figure 5A).

We then used in situ hybridization (RNAscope technology) to localize *auts2a* expression in the developing brain. At the beginning of late embryonic brain establishment (stage 29), *auts2a* was expressed in the forebrain (telencephalon and diencephalon), midbrain (optic tectum), hindbrain (cerebellum and rhombencephalon), and eyes (Figure 6). In the hindbrain, the expression was especially strong in the rhombencephalon, while *auts2a* could be detected at much lower levels in the cerebellum. In midbrain, strong *auts2a* levels were found in the optic tectum. Lower, yet significant *auts2a* levels were detected in the forebrain in the telencephalon and diencephalon. Finally, very low *auts2a* levels were detected in the eyes.

## 4. Discussion

### 4.1. Origins and Evolution of the auts2 Gene Repertoire

In the present study, we provide a comprehensive characterization of the evolution of the *auts2* gene repertoire in teleosts. Using a total of 76 species from the three main teleost clades (Elopomorpha, Osteoglossomorpha, and Clupeocephala), we provide comparative genomic data demonstrating that *auts2a* and *auts2b* originate from TGD. Our teleost-wide observations are fully consistent with observations previously made in zebrafish indicating that *auts2a* and *auts2b* originate from TGD [22]. After whole genome duplication, the loss of one ohnologous copy is the most common evolutionary fate with approximately only 12–24% genes retained in duplicates for protein-coding genes [34,35,41]. In the case of *auts2*, both copies were retained in Elopomorpha and Clupeocephala basal teleost lineages. Our analysis revealed a diversity of subsequent evolutionary fates leading to all possible retention patterns (one of the two genes or both) depending on the species. The *auts2a* gene was however retained in most lineages and appeared to be lost relatively recently, when lost. In Characiphysae (i.e., tetras, piranhas and catfishes), *auts2a* is present in the red-bellied piranha (*Pygocentrus nattereri*) and in the electric eel (*Electrophorus electricus*). In contrast, *auts2a* could not be identified in three species: Mexican tetra (both morphs), channel catfish, and iridescent shark catfish. The identification of possible *auts2a* gene remnants in conserved synteny blocks in these species would be in favor of the loss of *auts2a* in these species, even though the remaining amino acid sequences are very short (i.e., a few amino acids).

In contrast to *auts2a*, *auts2b* was lost in many species and several lineages, sometimes relatively early during evolution. This is, for instance, the case for the 52 species belonging to the Acanthomorphata lineage in which *auts2b* was systematically absent with a non-functional gene remnant being found in the corresponding ohnologous genomic region. Gene remnants were systematically identified in the eight species that were investigated (Appendix A). The *auts2b* gene was also absent from the genome of the three investigated Osteoglossomorpha species. This pattern is compatible with a rapid loss of *auts2b* after the Osteoglossomorpha radiation, even though we cannot rule out that this loss occurred later during evolution. It should however be stressed that we failed in identifying any *auts2b* gene remnant in these species. Finally, we also failed in identifying *auts2b* in European eel, even though the gene is present in the closely related European conger, which suggests a very recent gene loss.

After WGD, ohnologous gene copies can have different evolutionary fates. While the most common fate is the loss of one copy relatively rapidly after WGD (i.e., pseudogenization), both copies can remain and exhibit different behaviors [42,43]. Ohnologous copies can retain similar functions and act in a dose-dependent manner through cumulative effects, to produce sufficient amounts of the protein to perform the same ancestral function [42]. In contrast, the portioning of ancestral functions between the two ohnologous copies can occur in the case of sub-functionalization [43]. Finally, it is also possible that one copy retained the ancestral function, while the other acquired new functions (i.e., neo-functionalization). In the case of *auts2*, we observed that retention of a single copy, mostly *auts2a*, was the most common fate and was associated with strikingly similar expression patterns characterized by predominant expression in the embryo, adult brain, and gonads, which is also observed in spotted gar. These observations are consistent with the pseudogenization of one ohnologous copy, the remaining copy retaining ancestral functions. We, however, showed that the loss of one copy did not occur rapidly after TGD (i.e., before the divergence between Elopomorpha and Osteoglossomorpha/Clupeocephala), but later during evolution. In the case of Northern pike, the *auts2b* copy appears to be currently undergoing pseudogenization. Together, these observations suggest that retaining the two ohnologous copies, at least for a while, provided an evolutionary advantage. It should, however, be noted that most species eventually lost one of the two copies, suggesting that a single copy of the gene is sufficient. When looking at the *auts2* gene structure and predicted protein coding size and domains, we observed that in teleost fish (as highlighted here in medaka and zebrafish in Figure 4), the *auts2a* gene encodes for a long (1282 and 1278 amino acids in medaka and zebrafish, respectively) protein that is highly similar to the human AUTS2 protein [4]. The human gene is also known to encode many alternatively spliced variants. Among them, the best-documented ones encode short forms of 786 (S-AUTS2-Var1) and 711 (S-AUTS2-Var2) amino acids, both corresponding to the C-terminal part of the full-length protein and starting from exons 8 and 9, respectively [2,4]. The Auts2b protein described here in teleosts is also much shorter than the full-length Auts2a protein with 748 amino acids in zebrafish and roughly corresponds to mammalian S-AUTS2-Var1, both in terms of size and protein domain features (Figure 4). A similar size of the Auts2b protein was also observed in all teleost species that have retained *auts2b* (Appendix A). Data in mammals have suggested that the N-terminal and C-terminal regions of AUTS2 have different functions and are associated with different syndrome severity [4,6,14,15]. Recent data from a Yeast 2-hybrid screen using long and short isoforms suggest that the C-terminus of AUTS2, which is included in both isoforms, is responsible for transcription activation, whereas the N-terminal region, which is specific to the long isoform, is able to repress transcription [15]. In zebrafish, *auts2a* and *auts2b* exhibit distinct spatio-temporal expression patterns. The *auts2a* gene is maternally inherited and zygotically expressed, while *auts2b* is only expressed after zygotic genome activation during zebrafish development [22]. As in humans, several alternatively spliced transcripts are predicted for *auts2a* genes in teleosts both by Ensembl and by 5′ race experiments in zebrafish [22]. The ATG motif corresponding to the beginning of human AUTS2-C is also present in teleosts. Collectively, these data are consistent with different functions for full-length and short C-terminal proteins in vertebrates. While it is established that human AUTS2 can yield AUTS2 full-length and short C-terminal AUTS2 isoforms, different scenarios would be occurring in teleosts. Species in which only *auts2a* was retained would be similar to humans, with a single gene encoding both full-length and short C-terminal proteins. For species in which both *auts2a* and *auts2b* were retained, we speculate that the full-length form would be encoded by *auts2a*, while the short C-terminal forms would be encoded by *auts2a* and/or *auts2b*. This is also fully consistent with the loss of *auts2b* at different times during evolution in different lineages due to partial redundancy between Auts2b and the short C-terminal Auts2a. This raises the question of the loss of *auts2a* that we observed in three different species. It is possible that other proteins of the Auts2 family (i.e., Fbrs and Fbrsl1) have taken over the ancestral function of full-length Auts2a. This would be consistent with existing data in humans showing that FBRS and FBRSL1 are able to interact with PRC1 components [44]. In addition, FBRSL1 and AUTS2 exhibit a highly conserved domain, the AUTS2/FBRSL1 short homology (AHsh), of 46 aa in the N-terminal region of the protein [4]. We cannot however rule out that the absence of *auts2a* in these species is due to artifacts in genome assemblies, even though we have identified what could be gene remnants in corresponding ohnologous regions. Further investigations are needed to fully unravel this question.

### 4.2. A Strong Expression during Brain Formation

As shown here, both *auts2a* and *auts2b* are predominantly expressed in the adult brain in comparison to other organs and tissues in all investigated species with the exception of the bowfin. We also observed significant expression levels in whole embryos. During development, we observed a burst in *auts2a* expression at the time of neurula and brain regionalization, followed by a dramatic, but progressive increase during embryonic brain formation. These results are fully consistent with existing data in zebrafish in which *auts2a* mRNA could be detected using in situ hybridization from 10–11 h post-fertilization (hpf) (i.e., neurula) until 24 hpf (i.e., late embryonic formation) [22]. Existing data in zebrafish showed a wide expression in the brain of zebrafish at 24 hpf using digoxygenin probes and chromogenic detection [22]. Using RNAscope, a sensitive and quantitative technique [29,30,31] coupled with florescence detection, we were able to precisely map *auts2a* expression during late embryonic brain formation. In mammals, *Auts2* expression has been extensively studied in the developing brain. In postnatal and fully developed mouse brain, *Auts2* is expressed in glutamatergic projection neurons in the prefrontal cortex, pyramidal CA neurons in the hippocampus, granule neurons of the dentate gyrus, and Purkinje and Golgi cells in the cerebellum [12,18]. During late medaka embryonic brain formation (i.e., 34-somite stage), *auts2a* was expressed in the forebrain (telencephalon and diencephalon), midbrain (optic tectum), hindbrain (cerebellum and rhombencephalon), and in the eyes. To our knowledge, this is the first investigation of the spatial expression of the *auts2a* gene during medaka neurodevelopment. Two previous studies have, however, investigated *auts2a* expression in zebrafish and showed that *auts2a* was expressed in the forebrain, midbrain, and hindbrain of zebrafish at the 30-somite and 35-somite stages [6,23]. Another study showed that *auts2a* was expressed in the brain with stronger expression in the midbrain (optic tectum) of the zebrafish at the 30-somite stage [22]. The results obtained here in medaka at the 34-somite stage are therefore highly consistent with previous observations in zebrafish. In contrast to previous studies in fish, we were able to precisely map *auts2a* expression in the different regions of the late embryonic brain. In mice, *Auts2* is expressed in the cerebral cortex and cerebellum of the developing brain [12]. The *auts2a* expression pattern in medaka telencephalon and diencephalon reported here is therefore fully consistent with existing data in mice. Similarly, we were able to detect *auts2a* expression in the cerebellum, a highly conserved region between mammals and fish [45,46], at low, yet significant, levels. Together, these observations indicate a conserved expression in regions of the developing brain (i.e., developing cerebral cortex and cerebellum), which have been closely associated with the neuropathologies of autism [12] and with social behavior, memory, emotion, and learning in fish [47].

## 5. Conclusions

Here, we show that the whole genome duplication event that occurred at the stem of the teleost lineage yielded two ohnologous copies of the *auts2* gene, *auts2a* and *auts2b*. A comparative genomic analysis using a large number (*n* = 76) of species from the three main teleost clades (Elopomorpha, Osteoglossomorpha, and Clupeocephala) revealed different lineage-specific evolutionary fates. The retention of *auts2a* was observed in most investigated species with the exception of Mexican tetra (*Astyanax mexicanus*, both morphs), iridescent shark catfish (*Pangasianodon hypophthalmus*), and channel catfish (*Ictalurus punctatus*). In contrast, we were able to demonstrate that the *auts2b* gene was often lost during teleost evolution. This loss occurred either in the ancestor of large phylogenetic groups (e.g., Acanthomorphata, *n* = 52 species) or more recently in a species-specific manner. In most cases, the loss of *auts2b* was non-ambiguously supported by the presence of gene remnants in the corresponding conserved synteny blocs. Several species, including zebrafish, have retained the two gene copies, which were further duplicated in Salmonidae and Cyprininae. Despite different evolutionary fates, both genes exhibit a strikingly similar tissue distribution, which also appears to be conserved among vertebrates. The gene and protein structure of teleost *auts2a* is extremely conserved between teleost and mammals, both in terms of size and protein domains. We also observed that teleost *auts2b* genes encode for a shorter protein similar to a short protein isoform encoded by human and murine genes (S-AUTS2-Var1), both in size and in terms of conserved domains. We hypothesize that following genome duplication, *auts2a* retained the function of the full-length human AUTS2 protein, while *auts2a* and/or *auts2b* retained the function of the S-AUTS2-Var1 isoform, depending on the species. Using medaka, in which only *auts2a* was retained, we showed that *auts2a* messenger RNA was maternally inherited and expressed throughout embryonic development with bursts in expression during brain regionalization and late embryonic brain formation. We were also able to demonstrate that *auts2a* was expressed in the forebrain (telencephalon and diencephalon), midbrain (optic tectum), hindbrain (cerebellum and rhombencephalon), and eyes during late embryonic brain formation. These observations are consistent with expression patterns in mice in brain regions associated with neurodevelopmental disorders. Together, our observations show that *auts2* features and expression are highly conserved during vertebrate evolution. This sets the ground for the use of fish model species to decipher the role of *auts2a* during neurodevelopment.

## Figures and Tables

**Figure 1 cells-11-02694-f001:**
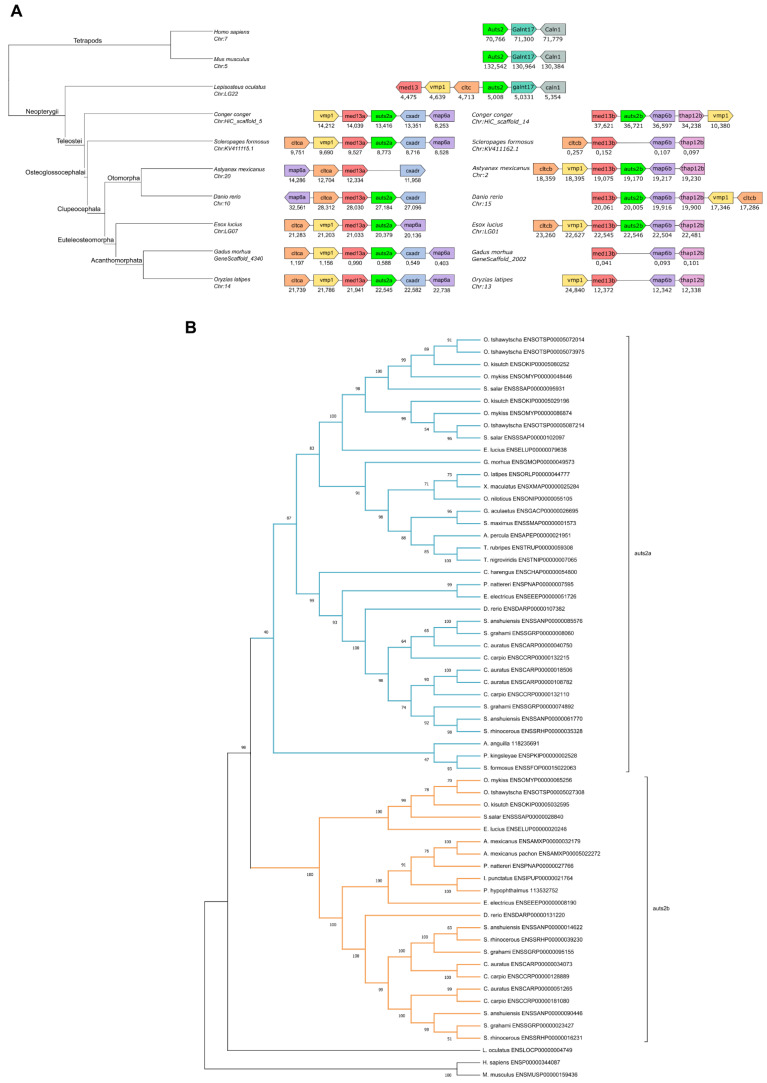
Identification of teleost *auts2* ohnologs and their origin using synteny and phylogenetic analysis (**A**). Identification of teleost *auts2a* and *auts2b* in 7 species using synteny analysis. Chromosome and scaffold numbers are indicated. Gene positions are in kilobases. The names of phylogenetic groups are indicated on tree branches. (**B**). Phylogenetic analysis of full-length Auts2 proteins using CLUSTALW alignment and maximum likelihood method. Protein accession numbers are provided in front of the species names. Bootstrap values are displayed on each node.

**Figure 2 cells-11-02694-f002:**
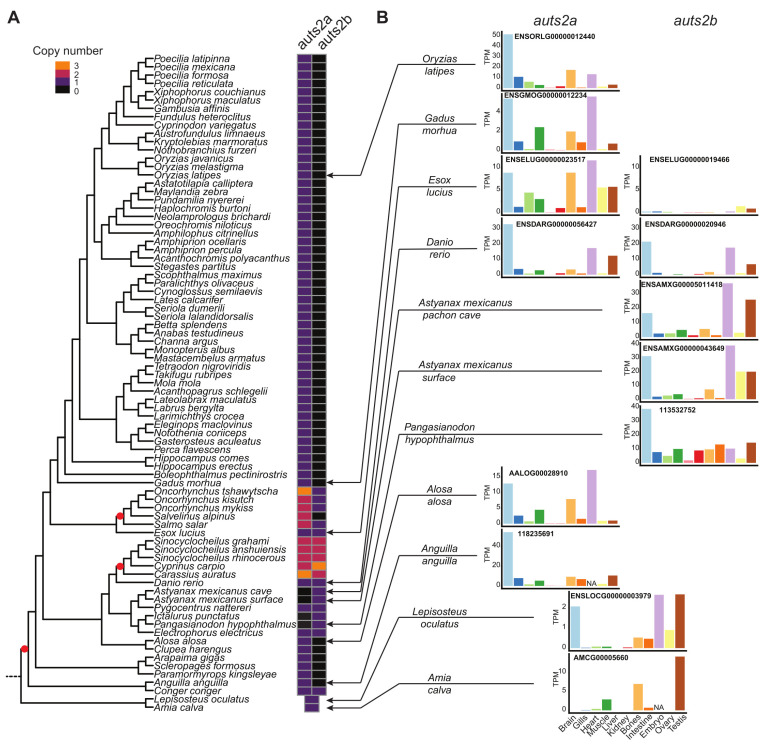
Comprehensive analysis of teleost *auts2* gene retention and expression after teleost-specific whole genome duplication. (**A**). Retention of teleost *auts2a* and *auts2b* in 78 species using synteny analysis. Whole genome duplication events are indicated by a red spot. (**B**). Tissue expression analysis of *auts2* using RNA-seq data in 10 species. Different colors correspond to different tissues.

**Figure 3 cells-11-02694-f003:**
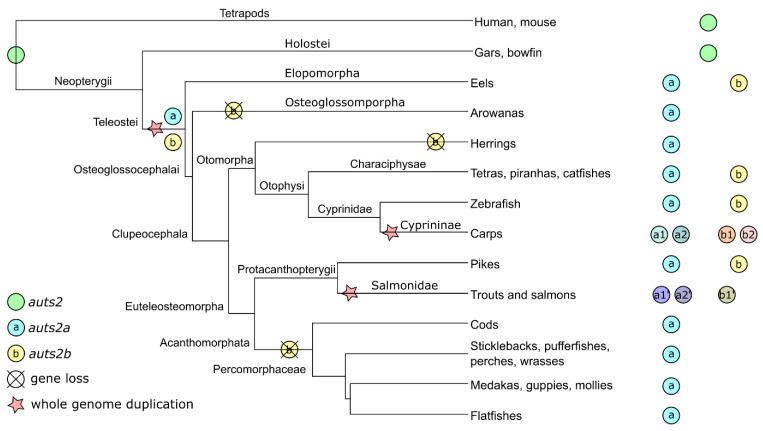
Teleost *auts2* ohnologs’ evolutionary history. The presence of a gene in a specific clade denotes the retention in at least one species of the clade.

**Figure 4 cells-11-02694-f004:**
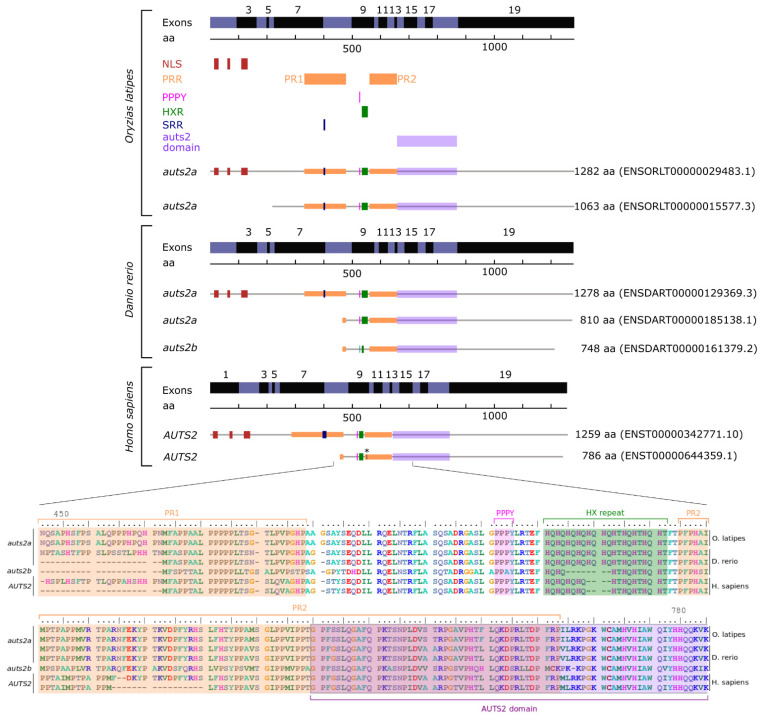
AUTS2 exons and protein domains in humans, zebrafish, and Japanese medaka based on Ensembl. NLS: nuclear localization sequence. PRR: proline-rich repeat domain. PPPY: PPPY protein binding motif. HXR: HX repeat motif. SRR: serine-rich repeat domain. * indicates the position of the initiation codon of the S-AUTS2-Var2 human short C-terminal isoform [4,6].

**Figure 5 cells-11-02694-f005:**
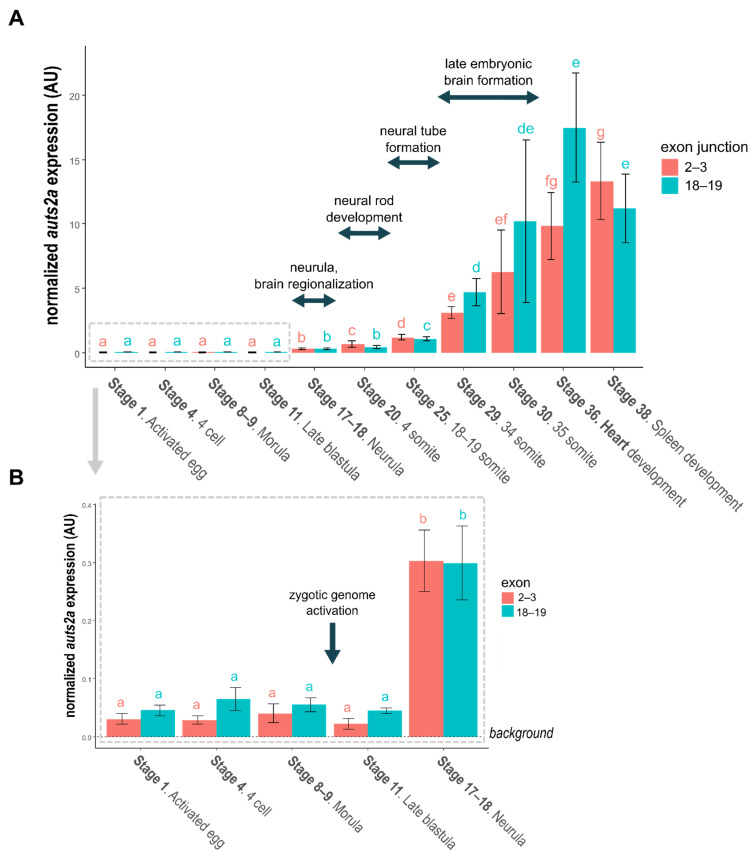
(**A**)**.** Japanese medaka *auts2a* expression during embryonic development. (**B**)**.** Focus on early embryonic development. For both panels bars represent the mean normalized expression calculated using 4 different pools of embryos. The standard deviation is displayed. Primers surrounding exons 2–3 junction target the full-length isoform. Primers surrounding the exons 18–19 junction target both full-length and short C-terminal isoforms. Comparisons were performed independently for each set of primer. Stages sharing the same letter are not significantly different (*p* < 0.05).

**Figure 6 cells-11-02694-f006:**
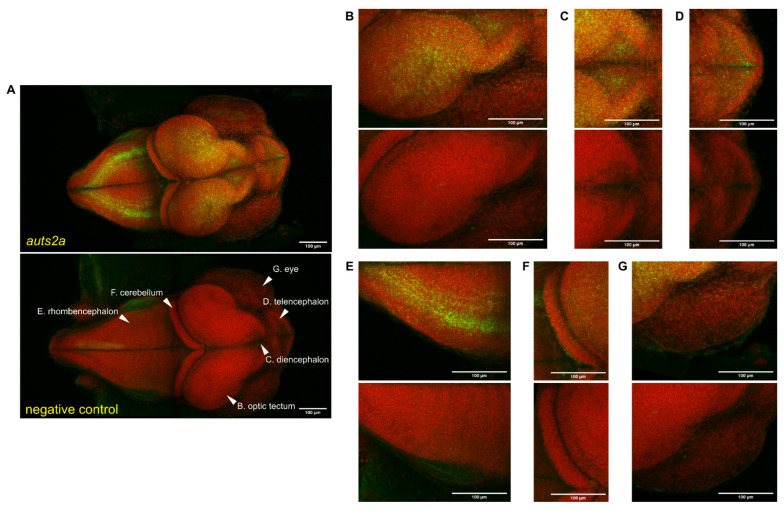
*auts2a* spatial expression in stage 29 medaka whole embryos using RNAscope. Green staining corresponds to the RNAscope signal. Red staining corresponds to the methyl-green-stained cell nucleus. An RNAscope probe targeting the *Bacillus subtilis dapB* gene (Accession #: EF191515) was used as a negative control. Images correspond to a maximum intensity Z axis projection for the RNAscope probe and standard deviation Z axis projection for methyl green. (**A**). Whole embryo. (**B**). Optic tectum. (**C**). Diencephalon. (**D**). Telencephalon. (**E**). Rhombencephalon. (**F**). Cerebellum. (**G**). Eyes.

## Data Availability

All data used are available either as Appendix A or in public repositories referenced in the Methods Section.

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
