# Peer review of "auts2 Features and Expression Are Highly Conserved during Evolution Despite Different Evolutionary Fates Following Whole Genome Duplication"

_cells, 2022, doi:10.3390/cells11172694_

Round 1

Reviewer 1 Report

In this manuscript, the authors comprehensively analyzed the evolution of the auts2 gene repertoire in teleosts using 74 species, including RNA-seq data and validation by in situ hybridization in Japanese medaka whole embryos. The study is interesting and examine all possibilities to answer the research question. Thus, this reviewer has only one minor comment. 

Lines 298 - 299: "In summary (Figure 3) the comprehensive gene analysis shows that, in contrast to many other genes [34]," Please provide the names of some exemplified genes.

Author Response

We thank reviewer 1 for their constructive comment. In the revised manuscript we have added 7 examples of genes for which one dupllicated copy was lost rapidly after the teleost-specific whole genome duplication. Corresponding references were added.

Reviewer 2 Report

In this paper, the authors describe the phylogenomic analysis, duplication and expression of the AUTS2 gene during embryonic development, and in formed organs such as the gonads and the brain. The authors use an already available and published database that contains transcriptomes from 11 species of fish and belonging 11 tissues, including the embryos.

Furthermore, the authors determined the expression of the auts2a variant gene during neural development in medaka using the qRTPCR method. The expression of this gene during late embryonic development, which coincides with the formation of the brain, was determined. The stage 29 of embryo development was used for in situ RNA hybridization and localization of this gene in the telencephalon and diencephalon, optic tectum, cerebellum and rhombencephalon. In the rhombencephalon, the expression of the auts2a gene is the most intense compared to other parts of the brain. The authors presented significant phylogenetic analyzes of the presence of two variants genes(auts2a and auts2b) as well as their expression in different tissues and organs. The presence of transcripts of this transcriptional regulator in certain organs points to its importance in the development and function of certain organs, which represents a further challenge in determining the function of this protein in the regulation of gene expression. Although the work does not bring significant knowledge about the function of this gene, the published results are an important contribution to further directions to understand more the biological function of this highly conserved protein (gene) in specific tissues and organs, including embryonic development.

Author Response

We thank reviewer 2 for their positive feedback on the manucript. The revised version was carefully edited and the English was review by a native English speaker.